# p66α Suppresses Breast Cancer Cell Growth and Migration by Acting as Co-Activator of p53

**DOI:** 10.3390/cells10123593

**Published:** 2021-12-20

**Authors:** Qun Zhang, Yihong Zhang, Jie Zhang, Dan Zhang, Mengying Li, Han Yan, Hui Zhang, Liwei Song, Jiamin Wang, Zhaoyuan Hou, Yunhai Yang, Xiuqun Zou

**Affiliations:** 1Hongqiao International Institute of Medicine, Tongren Hospital/Faculty of Basic Medicine, Shanghai Jiaotong University School of Medicine, Shanghai 200025, China; zhang.qun@sjtu.edu.cn (Q.Z.); Zhang_yihong2022@163.com (Y.Z.); zhangjie0307zjzz@sjtu.edu.cn (J.Z.); zhangdan_0117@163.com (D.Z.); limengying37@126.com (M.L.); Yanhan1108@126.com (H.Y.); zanghuier@outlook.com (H.Z.); jesuisjm@163.com (J.W.); houzy@sjtu.edu.cn (Z.H.); 2Shanghai Key Laboratory for Tumor Microenvironment and Inflammation, Department of Biochemistry & Molecular Cellular Biology, Shanghai Jiaotong University School of Medicine, Shanghai 200025, China; 3Shanghai Pulmonary Tumor Medical Center, Shanghai Chest Hospital, Shanghai 200025, China; songliwei_2011@163.com; 4Naruiboen Biomedical Technology Corporation Limited, Linyi 277700, China

**Keywords:** p66α, p53, cell growth, metastasis, migration, breast cancer

## Abstract

p66α is a GATA zinc finger domain-containing transcription factor that has been shown to be essential for gene silencing by participating in the NuRD complex. Several studies have suggested that p66α is a risk gene for a wide spectrum of diseases such as diabetes, schizophrenia, and breast cancer; however, its biological role has not been defined. Here, we report that p66α functions as a tumor suppressor to inhibit breast cancer cell growth and migration, evidenced by the fact that the depletion of p66α results in accelerated tumor growth and migration of breast cancer cells. Mechanistically, immunoprecipitation assays identify p66α as a p53-interacting protein that binds the DNA-binding domain of p53 molecule predominantly via its CR2 domain. Depletion of p66α in multiple breast cells results in decreased expression of p53 target genes, while over-expression of p66α results in increased expression of these target genes. Moreover, p66α promotes the transactivity of p53 by enhancing p53 binding at target promoters. Together, these findings demonstrate that p66α is a tumor suppressor by functioning as a co-activator of p53.

## 1. Introduction

p66α (GATA zinc finger domain-containing protein 2A, GATAD2A) and p66β (GATA zinc finger domain-containing protein 2B, GATAD2B) are two highly related 66-kDa proteins identified as MBD2 (methyl-CpG binding domain protein 2) interacting proteins in yeast two-hybrid screening assays [1]. They share two conserved domains: the N-terminal CR1 domain and the C-terminal GATA zinc finger-containing CR2 domain [1].

GATA zinc fingers comprise general configurations of Cys-X2-Cys-X17-Cys-X2-Cys and can specifically bind the consensus sequence A/T(GATA)A/G in target promoters to modulate gene transcription [2,3,4,5]. The three-dimensional structure of the zinc fingers in GATA-1 has revealed that the core DNA-binding domain contains two anti-parallel beta sheets and an alpha helix followed by a loop, which is the structural basis for interacting with DNA grooves [6]. The classical GATA transcription factors contain one or two conserved zinc fingers, and all vertebrate GATA transcription factors have two zinc fingers [5]. The C-terminal zinc finger specifically recognizes the consensus sequence in the target promoters, and the N-terminal zinc finger predominantly interacts with the partner proteins such as the FOG family proteins to stabilize the DNA-protein complex [7,8,9]. However, there is only one GATA zinc finger in p66α/p66β, which has not been demonstrated to be capable of binding DNA.

p66α/p66β were identified as essential components of the highly conserved, ATP-dependent, chromatin remodeling NuRD (Nucleosome Remodeling and Deacetylase) complex [10], which contains CpG-binding proteins MBD2/3, ATP-dependent remodeling enzymes CHD3/4, histone deacetylases HDAC1/2, p66α and p66β, histone chaperones RbAp46/48, and DNA-binding proteins MTA1/2/3 [11]. The NuRD complex dominates a transcriptional repression process of DNA methylation depending on its nucleosome remodeling activity and participates in various cellular functions such as DNA damage repairing and mRNA splicing [12,13,14,15,16,17,18,19,20]. p66α is shown to be necessary for targeting the NuRD complex to specific nuclear loci and mediating histone tail interaction [10,12]. The interaction between the p66α CR1 domain and MBD2 coiled-coiled domain is essential for the recruitment of CHD4 to the NuRD complex, and its deficiency would disrupt the repressive function of the complex [1,10,21]. ZMYND8 can recruit the NuRD complex to the DNA breaks via p66α and subsequently promote DNA repair by homologous recombination [22]. Moreover, phosphorylated p66α can recruit Suv39h1 and HDAC2 to establish repressive histone marks that affect the splicing process of *Neurexin-1* mRNA, leading to changed connectivity of the activated neurons [12]. Most recently, p66α mutants were determined to impair the recruitment of CHD4 to the MBD2-NuRD complex, leading to disturbed silence effects on γ-globin and an elevated fetal hemoglobin (HbF) level in patients with β-thalassemia [13,23]. Moreover, SUMO modification of p66α can enhance the transcriptional repression of the NuRD complex by promoting protein interactions within the complex [24]. Inhibition of p66α promotes iPSC reprogramming by disrupting Mbd3/NuRD repressive activity on the pluripotency circuitry [20]. Biologically, p66α is expressed during early embryonic development, and its genetic deletion leads to a lethal embryo phenotype in mice [25,26].

Studies have shown that p66α is involved in tumor development. For example, depletion of p66α results in suppressed cell proliferation in thyroid cancer cells [17], and p66α interacts with STAT3 to suppress its phosphorylation and K63 ubiquitination in myeloid-derived suppressor cell (MDSC), resulting in the reduced differentiation and the repressed immune suppressive function of MDSCs [27]. By genome-wide meta-analyses, p66α has been identified as a risk gene for breast cancer [28,29]. However, the precise role of p66α in breast cancer remains to be elucidated.

p53 is one of the most studied classical tumor suppressors [30,31,32]. When cells are insulted by stresses such as DNA damage, hypoxia, and oncogene activation, p53 is activated to transcriptionally regulate genes involved in the cell cycle, cell death, cell migration, and angiogenesis, which contribute to the maintenance of genomic stability, suppression of cell growth and tumor development [33,34,35,36,37]. However, *TP53* is one of the most altered genes in cancer and is mutated in about 23% of breast cancer samples [38]. The DNA-binding domains of p53 are a hot spot bearing many point mutations [39]. Some point mutations, such as mutants at amino residues R248, R273, and R280, have been classified as “contact mutants” since they directly impact contact between p53 and the target DNA sequences [38]. However, some p53 mutants still retain function as sequence-specific transcription factors, showing an altered but not totally eliminated ability to transactivate target promoters [40,41,42]. Thus, a single residue alteration in p53 can trigger a considerable diversity in the response profile of different response elements [40,43,44,45,46,47].

In this study, we report that p66α suppresses the growth and migration of breast cancer cells by interacting with p53 to enhance the binding activity of p53 at target promoters and increase the expression of p53 target genes. Moreover, p66α can also promote the transcriptional activation function of the p53 R280K found in MDA-MB-231 cells.

## 2. Materials and Methods

### 2.1. Plasmids

The pcDNA-Flag-p66α plasmid was kindly gifted by Dr. Reiner Renkawitz, Genetisches Institut Justus-Liebig-Universitaet. The HA-p66α and pCDH-Flag-p66α plasmids were subcloned into pCMV5-HA-vetcor and pCDH-CMV-MCS-EF1-Puro-vector, respectively. The deletion mutant plasmids Flag-p66α-CR1 and Flag-p66α-CR2 were subcloned into pCMV4-Flag-vector by PCR method between *Bam*H I and *Eco*R I restriction enzyme sites. PCR amplification was carried out using Phanta Max Super-Fidelity Polymerase (P505-d1, Vazyme, Piscataway, NJ, USA). The GFP-p53 plasmid was kindly gifted by Pf. Jing Yi, Shanghai Jiaotong University School of Medicine. The Flag-p53 and its truncation mutant plasmids were subcloned into pCDNA3.1-Flag-vector and pGEX-4T1-vector by PCR method between *Bam*H I and *Eco*R I restriction enzyme sites. pLKO.1-p66α shRNA plasmids were generated with the oligonucleotides 5′-TTCTCTCAGAATGTCTGCCGG-3′ and 5′-CCGGCAGACATTCTGAGAGAA-3′, which target the coding region of the p66α transcript. The promoter DNA sequences of human *BAX* (−478 to +12 bp) and *NOXA* (−471 to −1 bp) were subcloned into pGL3-basic-vector to create the corresponding luciferase reporters.

### 2.2. Cell Culture, Transfections, and Viral Infection

HEK-293T, MDA-MB-231, MCF-7 cells were cultured in Dulbecco’s modified Eagle’s medium (DMEM) supplemented with 10% fetal bovine serum (FBS), penicillin (50 U/mL)/streptomycin (50 μg/mL). MCF-10A cells were cultured in Dulbecco’s Modified Eagle Medium/Nutrient Mixture F-12 supplemented with 5% horse serum, EGF (20 ng/mL), insulin (10 μg/mL), hydrocortisone (0.5 μg/mL), cholera toxin (100 ng/mL), and penicillin (50 U/mL)/streptomycin (50 μg/mL). All of the cells were cultured at 37 °C under 5% CO2 in a humidified chamber.

For transient transfection, plasmids were transiently transfected into cells at approximately 70% confluence using PEI reagent (#23966-2, polysciences, Warrington, PA, USA) according to the manufacturer’s instructions. After 48 h of transfection, cells were harvested.

The stable cells were established via the viral infection method. For lentivirus packages, indicated plasmids were transfected into HEK-293T cells according to the transient transfection method. The supernatants containing viruses were collected at 48 h and 72 h after transfection and used to infect target cells. Twenty-four hours post-infection, puromycin (1 μg/mL for MDA-MB-231 and MCF-10A cells, 2 μg/mL for MCF-7 cells) was added to select the stable cells for 5 days.

### 2.3. Western Blot, Immunofluorescence, Co-IP and GST Pull Down

Western blotting and immunofluorescence assays were carried out as described [48,49]. Plasmids encoding Flag-p66α, GFP-p53 or HA-p66α, Flag-p53 were transiently transfected into HEK-293T cells, and 24 h after transfection, cells were lysed in the cell lysis buffer containing 20 mM Tris (pH 7.5), 150 mM NaCl, 2.5 mM EDTA, DTT, and protease inhibitor mixture. 

GST-p53 and its truncation mutant proteins were expressed in *E. coli* BL21 and purified by using GST beads (17-0756-01, GE Healthcare, Little Chalfont, Buckinghamshire, UK).

The antibodies used were as follows: mouse anti-Flag antibody (F3165, Sigma-Aldrich, St. Louis, MO, USA), rabbit anti-Flag antibody (F7425, Sigma-Aldrich, St. Louis, MO, USA), rabbit anti-GST antibody(#2622, Cell Signaling Technology, Danvers, MA, USA), mouse anti-HA antibody (901503, BioLegend, San Diego, CA, USA), rabbit anti-GATAD2A (H-162) antibody (sc-134712, Santa Cruz Biotechnology, Santa Cruz, CA, USA), mouse anti-p53 (DO-1) antibody (sc-126, Santa Cruz Biotechnology, Santa Cruz, CA, USA), rabbit anti-p53 (7F5) antibody (#2527, Cell Signaling Technology, Danvers, MA, USA), anti-β-actin antibody (66009-1-Ig, Proteintech, Chicago, IL, USA), mouse GFP-antibody (50430-2-AP, Proteintech, Chicago, IL, USA), Alexa Fluor 488 Donkey anti-Mouse IgG (A-21202, Invitrogen, Carlsbad, CA, USA), Alexa Fluor 568 Donkey anti-Rabbit IgG (A-10042, Invitrogen, Carlsbad, CA, USA), Alexa Fluor 568 Rabbit anti-Mouse IgG (A-11061, Invitrogen, Carlsbad, CA, USA).

### 2.4. Reverse Transcription PCR (RT-PCR) and Quantitative Real-Time PCR (qRT-PCR)

Total RNA from stable cells was extracted using the TRIzol reagent (15596026, Life Technologies, Carlsbad, CA, USA) according to the manufacturer’s instructions. Reverse transcription procedure was described previously; three micrograms of the treated total RNA were used for cDNA synthesis in a 20 μL reaction using SuperScript II reverse transcriptase (18064022, Invitrogen, Carlsbad, CA, USA) [48,49,50]. 

qRT-PCR was performed using Hieff SYBR Green Master Mix (11201ES03, Yeasen Biotech, Shanghai, China), and reactions were performed on Roche LightCycler^®^ 480 II Real-Time PCR System. Data were acquired and analyzed using the 2^−ΔΔCT^ method with ACTIN as an endogenous control. Probes and primers used in qRT-PCR are listed in Appendix A.

### 2.5. Chromatin Immunoprecipitation (ChIP)

ChIP assays were carried out in MDA-MB-231-Luc shVector and shp66α cells. The cells were grown in 100 mm plates to 90% confluence and fixed by 1% formaldehyde for 10 min in the cell culture incubator. The cross-linking reaction was stopped by 0.125 M glycine in 1 × PBS at room temperature for 5 min. Cells were harvested, washed, and lysed in the ChIP lysis buffer containing 150 mM NaCl, 50 mM Tris-HCl (pH7.5), 5 mM EDTA, NP-40 (0.5% vol/vol), TritonX-100 (1.0% vol/vol). The chromatin was sonicated to fragments ranging from 300-800 bp in size. Immunoprecipitation was performed using the antibody against p53 (mouse, DO-1, Santa Cruz Biotechnology, Santa Cruz, CA, USA), mouse normal IgG (sc-2025, Santa Cruz Biotechnology, Santa Cruz, CA, USA). The immunoprecipitated DNAs were amplified by qRT-PCR. Primers used for ChIP qRT-PCR were listed in Appendix A.

### 2.6. Luciferase Reporter Assay

Luciferase reporter assay was performed in HEK-293T cells. After 24 h transfection, cells were harvested in luciferase lysis buffer. Firefly luciferase activity (MA0519, Meilunbio, Dalian, China) and β-galactosidase activity (631712, Takara, Tokyo, Japan) were measured using the Commercial reagents according to the manufacturer’s instructions.

### 2.7. Cell Counting Assay

Cells (3000 per well) were plated in a 96-well plate containing 100 μL/well medium, and cell counting assays were performed according to the manufacturer’s instructions of Cell Counting Kit-8 (CK04, Dojindo, Shanghai, China). Absorbance at 450 nm was quantified at different time points using the Multiskan MK3 microplate reader (Thermo Fisher Scientific, Waltham, MA, USA). The cell growth curve was drawn with the Graphpad Prism.

### 2.8. Cell Migration Assay

Migration assays were performed as described previously [50]. After being starved for about 12 h, stable cells were harvested using trypsin and counted. A total of 5 × 10^4^ cells were resuspended in the serum-free medium and placed in the upper 8 μm pore transwell filters (3495, Corning Incorporated, Corning, NY, USA ). Complete DMEM medium was added to the bottom chamber as attractants. The chamber was incubated at 37 °C and in a 5% CO_2_ atmosphere for 18 hours. The cells were fixed with 4% poly-formaldehyde for 15 min, and cells were stained with Coomassie Brilliant Blue (G250) for 30 min. Non-migrated cells on the upper side of the filter were removed with a cotton swab. The chamber was washed with PBS 2–3 times and dried up at 37 °C [50]. The migrated cells on the bottom of filter were quantified by counting nine randomly chosen fields (20×) using microscopy in each well. Experiments were repeated in triplicates.

### 2.9. Immunoprecipitation Mass Spectrometry

A total of 4 × 10^8^ cells were lysed in lysis containing 20 mM Tris-HCl (pH 8.0), 150 mM NaCl, 2.5 mM EDTA, 0.5% NP-40, 0.2 mM PMSF, and 0.1 mM cocktail. Cell lysates were precleared with the protein A-agarose beads (sc-2003, Santa Cruz Biotechnology, Santa Cruz, CA, USA) for 2 h and then incubated with the anti-Flag agarose M2 beads (F2426, Sigma-Aldrich, St. Louis, MO, USA) at 0.5 mL of beads per 100 mg of cell lysate for 2 h to overnight with rotation. The M2 beads were washed four times with buffer BC500 containing 20 mM Tris-HCl (pH 7.8), 500 mM KCl, 0.2 mM EDTA, 10% glycerol, 10 mM mercaptoethanol, 0.2% NP-40, 0.2 mM PMSF, and 0.1 mM cocktail. The protein complex was eluted with the Flag peptides (Sigma-Aldrich, St. Louis, MO, USA) at 0.2 mg/mL in buffer TBS containing 20 mM Tris-HCl (pH 7.8), 50 mM NaCl,0.2 mM PMSF, and 0.1 mM cocktail. The eluted proteins were resolved on 4 to 12% sodium dodecyl sulfate-polyacrylamide gel electrophoresis gels for Western blotting and silver and colloidal staining analyses. The proteins located between 40-kDa and 55-kDa were excised from the gels and identified by standard mass spectrometry.

### 2.10. Subcutaneous Xenograft Assay

Female BALB/c nude mice were purchased from SLAC Laboratory Co. Ltd. (Shanghai, China) and divided into two groups randomly. The total number of 2 × 10^6^ (in 100 μL of PBS per mouse) luciferase-expressing MDA-MB-231 cells with or without p66α depletion cells was injected into 6-week-old nude mice subcutaneously per group (*n* = 10). Three weeks after inoculation, the mice were euthanized and the tumors were dissected and fixed in 4% paraformaldehyde. The volume of tumors was estimated according to the formula V = 0.52 × L × W^2^.

### 2.11. Lung Metastasis Assay and Hematoxylin and Eosin (H&E) Staining

Female BALB/c nude mice were purchased from SLAC Laboratory Co. Ltd. (Shanghai, China) and divided into two groups randomly. To analyze lung metastasis, a total number of 2 × 10^6^ (in 100 μL of PBS per mouse) luciferase-expressing MDA-MB-231 cells with or without p66α depletion cells were injected into the tail vein of 6-week-old female mice (*n* = 10 for each group). After inoculation, bioluminescence imaging of metastatic tumors was carried out using Xenogen IVIS Imaging System every two weeks. 

For H&E staining, the mice were euthanized and the lungs were dissected and fixed in 4% paraformaldehyde for 48 h. Samples were dehydrated in graded ethanol series followed by xylene. Subsequently, samples were then embedded in paraffin, sectioned onto slides (5 mm thick), and dried. Slides were deparaffinized using the standard procedure (xylene × 2 washes, 100% ethanol × 2 washes, 95% ethanol, 70% ethanol, 50% ethanol) and stained with H&E (HT110132, Sigma-Aldrich, St. Louis, MO, USA).

### 2.12. Survival Analysis and Correlation Analysis

The survival analyses of p66α and p53 were generated from Kaplan-Meier plots online tools, which are capable of assessing the effect of 54 k genes (mRNA, miRNA, protein) on survival in 21 cancer types including breast cancer. The patient samples are split into two groups according to the expression level of p66α or p53. The two patient cohorts are compared by the Kaplan-Meier survival plot, and the hazard ratio with 95% confidence intervals and log-rank *p* value is calculated. (http://kmplot.com, accessed on 3 December 2021)

The correlation analysis of p66α and p53 in breast cancer was generated with the online tools based on the TCGA database GEPIA (http://gepia.cancer-pku.cn/, accessed on 3 December 2021).

### 2.13. Statistical Analysis

Data presented as mean ± SD were analyzed by the independent Student’s *t* test.

## 3. Results

### 3.1. p66α Inhibits Breast Cancer Cell Growth

To examine if p66α participates in breast-cancer-relevant biological processes, we first performed extensive bioinformatics analyses to determine if there is a correlation between p66α expression level and the clinical outcome in breast cancer patients. We found that patients with the high level of p66α expression showed a better overall survival outcome (Figure 1A). This observation prompted us to examine the effect of p66α on breast cancer cell growth. First, p66α was depleted in MCF-7 cells by using specific targeting shRNAs (Figure 1B), and the cell growth rate was examined by CCK8 assays. Notably, depletion of p66α increased the growth of MCF-7 cells significantly (Figure 1C). Conversely, ectopic expression of p66α inhibited the growth of MCF-7 cells (Figure 1D,E). Similarly, knockdown of p66α in MDA-MB-231 resulted in increased cell growth (Figure 1F,G), while overexpression of p66α significantly inhibited cell growth (Figure 1H,I). Similar results were also observed in MCF-10A, which is classified as a normal breast cell line (Appendix A). Next, we subcutaneously injected the MDA-MB-231-Luc-shp66α or MDA-MB-231-Luc-shVector cells into nude mice, which were euthanized to dissect the tumors 18 days post-injection. The tumor sizes were measured and showed that the volume of the tumors in the knockdown p66α group was remarkably larger than those in the control group (Figure 1J,K), indicating that depletion of p66α enhances the tumor cell growth.

### 3.2. p66α Suppresses Breast Cancer Cell Migration and Metastasis

To determine the role of p66α in breast cancer cell migration, cells stably expressing shp66α or vector were subjected to transwell assays. Depletion of p66α expression markedly promoted cell migration in MCF-7 and MDA-MB-231 cells, respectively (Figure 2A,B,E,F). Conversely, ectopic expression of p66α resulted in decreased cell migration in MCF-7 and MDA-MB-231 cells (Figure 2C,D,G,H). Consistent with these observations, over-expression of p66α significantly inhibited cell migration in MCF-10A cells (Appendix A). To expand these observations, we used the tail vein injection lung metastasis model to examine the impact of p66α on tumor metastasis in vivo. MDA-MB-231-Luc-shp66α cells and control cells were injected into the tail vein of the nude mice, and metastatic tumor nodules were recorded by in vivo imaging technology every other week (Figure 2I). The luciferase signal intensities of lung metastatic nodules were significantly increased in the shp66α group compared with those in the control group (Figure 2J), and H&E staining assays revealed that the metastatic foci in the shp66α group were dramatically increased in tissue sections of the lungs (Figure 2K,L). Taken together, these observations demonstrate that p66α suppresses the migration and metastasis of breast cancer cells.

### 3.3. p66α Interacts with p53

To investigate how p66α elicits the suppressive effects on cell growth and migration, we performed an immunoprecipitation coupled mass spectrometry assay to explore p66α-interacting proteins. We established HEK-293T cells stably expressing Flag- p66α and performed affinity purification with Flag antibody. p53 was identified as a potential p66α-interacting protein. To confirm the interaction between p66α and p53, we first transiently co-expressed full-length p66α and p53 in HEK-293T cells, and co-immunoprecipitations were performed. Indeed, p66α robustly and reciprocally interacted with p53 in HEK-293T cells (Figure 3A,B). To further confirm that a p66α-p53 interaction occurs within the endogenous proteins, whole-cell lysates prepared from HEK-293T-Flag- p66α cells were used for immunoprecipitation with antibody against p66α, and Western blotting assays showed that p66α could co-elute with endogenous p53 protein (Figure 3C). Furthermore, indirect immunofluorescence staining showed that the two proteins co-existed in the nucleoplasm (Figure 3D). Collectively, these data suggest that p53 is a p66α-interacting protein.

### 3.4. The CR2 Domain of p66α and the DNA-Binding Domain of p53 Are Essential for the Interaction

To identify regions in p66α that are critical for the interaction between p66α and p53, p66α full-length or truncations were transiently co-expressed with GFP-p53 in HEK-293T cells. Co-IP assays demonstrated that CR1 only weakly bound p53, while the CR2 region robustly bound p53 (Figure 3E,F).

To determine the domains in p53 protein required for interacting with p66α, the full length of GST-p53 and three truncated p53 proteins were purified and incubated with HA- p66α protein prepared from HEK-293T cells. The co-precipitation experiments were performed by using GST beads, and the co-eluted HA- p66α protein was examined by Western blot assays by using HA antibody. The truncation mutant consists of 100-291aa containing DNA-binding domain motif was capable of binding p66α, which was comparable to that of the full-length p66α, while the mutants comprising 1 -99aa or 292- 393aa of p53 failed to bind p66α (Figure 3G,H). Taken together, the CR2 domain of p66α and the DNA-binding domain (100 -291aa) of p53 are essential for the interaction.

### 3.5. p66α Promotes the Expression of p53 Target Genes

To determine if the p66α-p53 interaction impacts p53 to regulate its target genes, we examined the expression of the known p53 target genes that control cell growth such as *BAX*, *GADD45A*, *NOXA*, and that control cell migration such as *PAI-1* in the MCF-7-shp66α or vector control cells by qRT-PCR. Notably, depletion of p66α decreased the expression of p53 target genes in MCF-7 cells (Figure 4A,B). Moreover, the increase in p66α levels in MCF-7 cells promoted the expression of these targets (Figure 4C,D), which is dependent on the expression of p53 (Figure 4E,F). A similar effect of p66α levels on the expression of the p53 target genes was observed in MDA-MB-231 cells (Figure 4G–L).

### 3.6. p66α Promotes the Transactivity of p53 by Enhancing p53 Binding at Target Promoters

To verify the effects of p66α on the transactivation ability of p53WT and p53R280K, we performed luciferase reporter assays with *BAX*-Luc or *NOXA*-Luc, which contained p53 responsive elements, respectively. Notably, both p53WT and p53R280K can transactivate *BAX* or *NOXA*, and a combination with p66α markedly increased *BAX-* or *NOXA*-driven reporter activities (Figure 5A,B).

To examine whether p66α affects the expression of p53, we detected the expression level of p53 in stable cells with knockdown or overexpression of p66α. The results showed that p66α did not affect the mRNA level of p53 (Appendix AA–C), nor did it affect the protein level of p53 (Appendix AD–F).

To determine the effects of p66α on the DNA-binding activity of p53, we performed ChIP assays in MDA-MB-231-shVector and MDA-MB-231-shp66α cells using antibodies specific to p53, and the enriched target DNA fragments were amplified by qPCR using primer sets flanking the functional p53 binding sites within target promoters. Depletion of p66α significantly decreased the binding of p53 to target promoters including *BAX*, *NOXA*, *GADD45A*, and *PAI-1* (Figure 5C). Taken together, these results indicate that p66α promotes the transactivity of p53 by enhancing p53 binding at target promoters.

### 3.7. High Expression of p66α and p53 Is Positively Correlated and Predicts Good Prognosis in Breast Cancer

To determine the clinical relevance of p66α and p53, we analyzed a Kaplan-Meier plotter database and found that high expression of p53 also predicted good prognoses in patients with breast tumors (Figure 6A). Moreover, the expression of p66α and p53 have a significant correlation in breast cancer samples (Figure 6B).

## 4. Discussion

### 4.1. p66α Functions as a Tumor Suppressor of Breast Cancer Cells

Breast cancer is one of the most common cancers in women, which has a mortality-to-incidence ratio of 15% and alone accounts for 30% of female cancers [51]. As a heterogenous malignancy, it could be divided into three categories according to molecular features, including hormone-positive subtype expressing estrogen receptor (ER+) or progesterone receptor (PR+), HER2-positive subtype expressing human epidermal receptor 2 (HER2+), and triple-negative breast cancer (TNBC) (ER-, PR-, HER2-) [52]. The TNBC treatment is the most challenging within breast cancer for its higher aggressive feature and lack of molecular targets [53,54,55], although there are a large number of tumor suppressor genes encoding proteins that inhibit cell transformation in breast cancer, such as *TP53*, *Rb1*, *PTEN*, *BRCA1/2*, *CDKN1B*, *CDKN2A,* etc. 

In this study, we demonstrate that p66α functions as a tumor suppressor that can inhibit breast cancer cell growth and migration. Expression of p66α can inhibit growth and migration of breast cancer cells MCF-7 and MDA-MB-231, as well as normal breast cells MCF-10A. Mechanistically, p66α functions as a novel co-activator of p53 to induce its target gene expression, and p66α promotes the transactivity of p53 by enhancing p53 binding at target promoters such as *BAX*, *NOXA*, *GADD45A*, and *PAI-1*, resulting in inhibition of cell growth and migration. Moreover, breast cancer patients having a higher level of p66α exhibit better prognosis. Taken together, our findings suggest that p66α could be a target for new strategies in the treatment of breast cancers.

### 4.2. p66α Functions as a Potent Co-Activator of p53 by Enhancing Its DNA-Binding Activity

Although p66α harbors a GATA motif within its CR2 region, there is no evidence to show that p66α can directly bind DNA sequences; rather, p66α acts like an adapter or a scaffold to participate in assembly of various protein complexes. Thus, the molecular mechanism by which p66α enhances the DNA-binding and transcriptional activity of p53 remains elusive and definitely needs to be explored further. For example, identifying the protein partners that associate with p53/p66α complex will provide new clues.

Another interesting observation is that p66α can enhance p53 target gene expression in MDA-MB-231 cells in a p53-dependent manner, since the p53 protein in MDA-MB-231 cells harbor the R280K mutation. R280K resides in the DNA-binding domain and results in impaired DNA-binding activity of p53 [56,57]. We found the p53R280K is also enriched at the promoters of target genes including *BAX*, *NOXA*, *GADD45A*, and *PAI-1* in MDA-MB-231 cells. Moreover, the luciferase reporter assays on promoters of *BAX* and *NOXA* showed that both p53 WT and R280K alone can induce their promoter activities, and adding p66α can further induce these promoter-luc activities.

We reasoned that this may be explained by at least two possible mechanisms: (1) p53R280K mutant still retains DNA-binding activity. Malcikova reported that p53R280K lost most of its binding ability to the *CDKN1A* and retained about 35% binding ability to the *BAX* gene [58]. In another study, the DNA-binding ability of p53R280K to the *GADD45A* gene was reduced, but not completely lost [59]. (2) MDA-MB-231 cells express a relatively higher level of p66α protein compared to MCF-7 and MCF-10A cells. Our data show that p66α can enhance the DNA-binding activity of p53 to its target promoters. The binding activities of p53R280K to its target promoters in MDA-MB-231 cells may be attributed to p66α, which may overcome the effect of the mutation.

## Figures and Tables

**Figure 1 cells-10-03593-f001:**
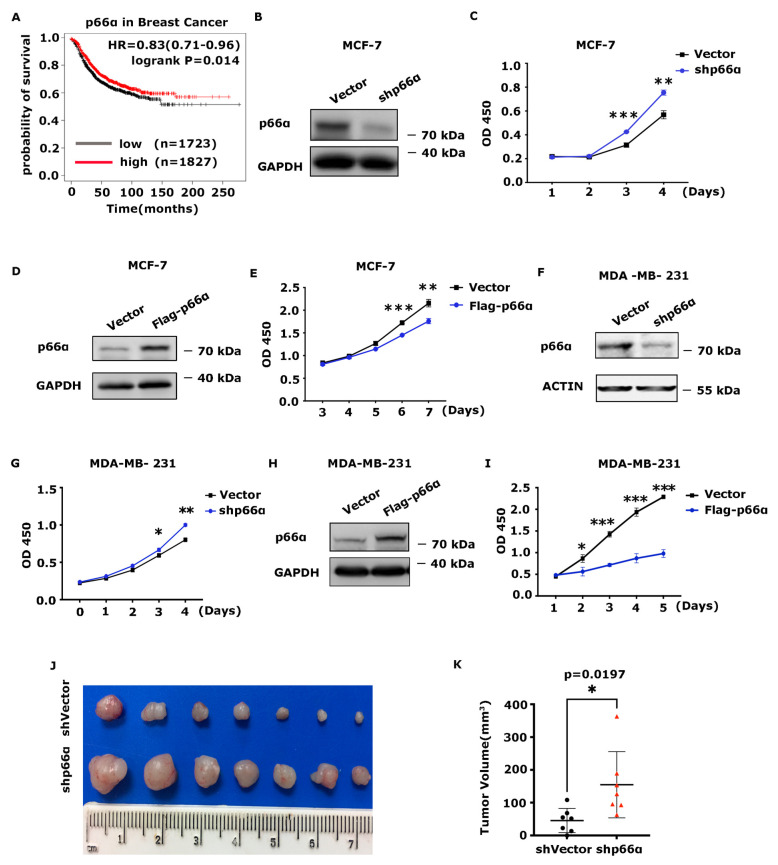
p66α inhibits breast cancer cell growth. (**A**) Kaplan-Meier plot of the relapse-free survival of patients with breast cancer stratified by p66α. (**B**) Western blot assays showed that p66α was effectively depleted in the MCF-7 cell line. (**C**) Cell growth curves of p66α knockdown MCF-7 cells and control cells. (**D**) Western blot assays showed that p66α was over-expressed in the MCF-7 cells. (**E**) Cell growth curves of MCF-7 cells overexpressing p66α or mock vector. (**F**) Western blot assays showed the depletion of p66α in the MDA-MB-231 cells. (**G**) The cell growth curves of p66α knockdown MDA-MB-231 cells and mock cells. (**H**) Western blot assays showed the expression of p66α in MDA-MB-231 cells. (**I**) The cell growth curves of MDA-MB-231 cells overexpressing p66α or mock control. (**J**) Tumor size of subcutaneous tumor formation assay. Knockdown of p66α in MDA-MB-231 cells promoted cell growth in vivo (*n* = 7). (**K**) statistical results of (**J**). * *p* < 0.05, ** *p* < 0.01, *** *p* < 0.001.

**Figure 2 cells-10-03593-f002:**
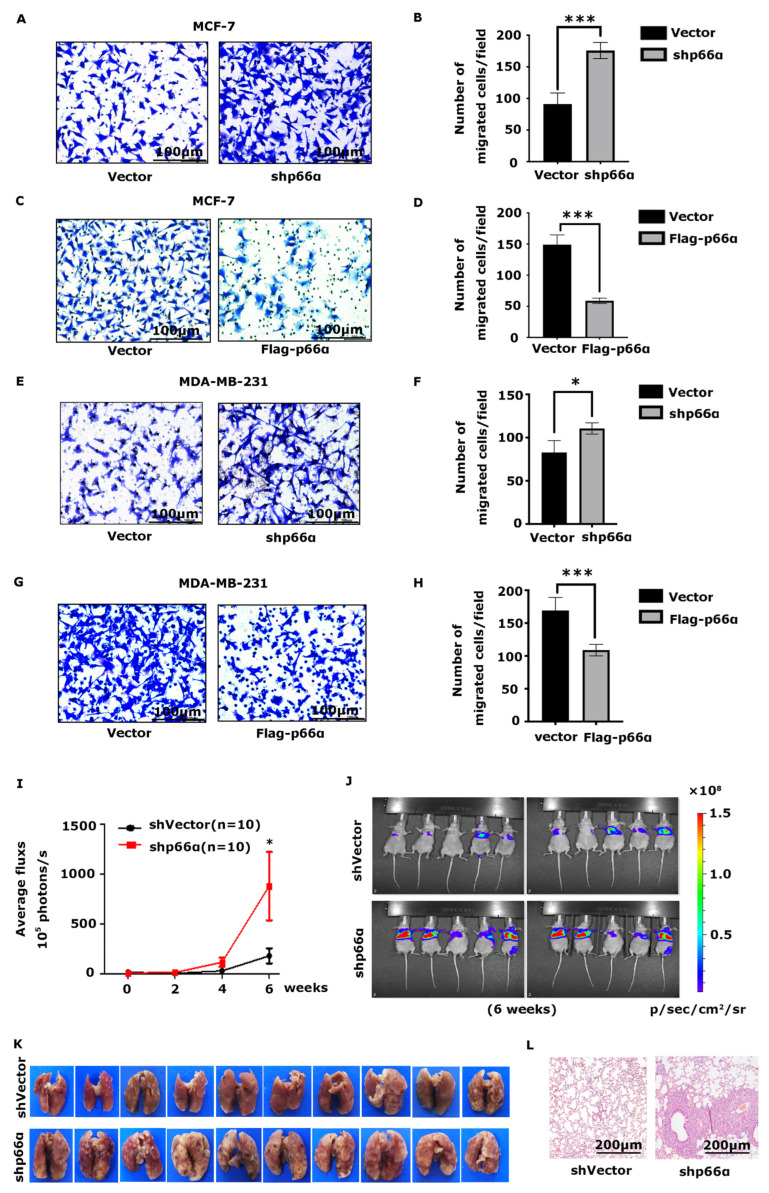
p66α suppresses breast cancer cell migration and metastasis. (**A**) Transwell assays demonstrated that knockdown p66α increased migration in MCF-7 cells (20×, scale bar: 100 μm). (**B**) Quantification analyses of cell migration abilities of p66α knockdown or vector MCF-7 cells. *** *p* < 0.001. (**C**) Transwell assays showed that an increase in p66α levels reduced migration in MCF-7 cells (20×, scale bar: 100 μm). (**D**) Quantification analyses of cell migration abilities of MCF-7 cells overexpressing p66α or vector. *** *p* < 0.001. (**E**) Depletion of p66α increased cell migration in MDA-MB-231 cells. (20×, scale bar: 100 μm). (**F**) Quantification analyses of cell migration abilities of p66α knockdown MDA-MB-231 cells. * *p* < 0.05. (**G**) Transwell assays showed that an increase in p66α levels reduced MDA-MB-231 cells migration (20×, scale bar: 100 μm). (**H**) Quantification analyses of cell migration abilities of MDA-MB-231 cells overexpressing p66α. *** *p* < 0.001. (**I**) MDA-MB-231-Luc cells stably expressing control vector or p66α shRNA were injected into the tail vein of female nude mice. Luciferase signal intensities of lung metastases in each group (*n* = 10) were recorded every two weeks. * *p* < 0.05. (**J**) Representative luciferase signal images of lung metastases at the sixth week. (**K**) Representative images of metastatic nodules in the lung. (**L**) H&E staining of lung tissue in nude mice, which were injected cells as indicated. The metastatic foci in the shp66α group were dramatically increased compared to the control group (10×, scale bar: 200 μm).

**Figure 3 cells-10-03593-f003:**
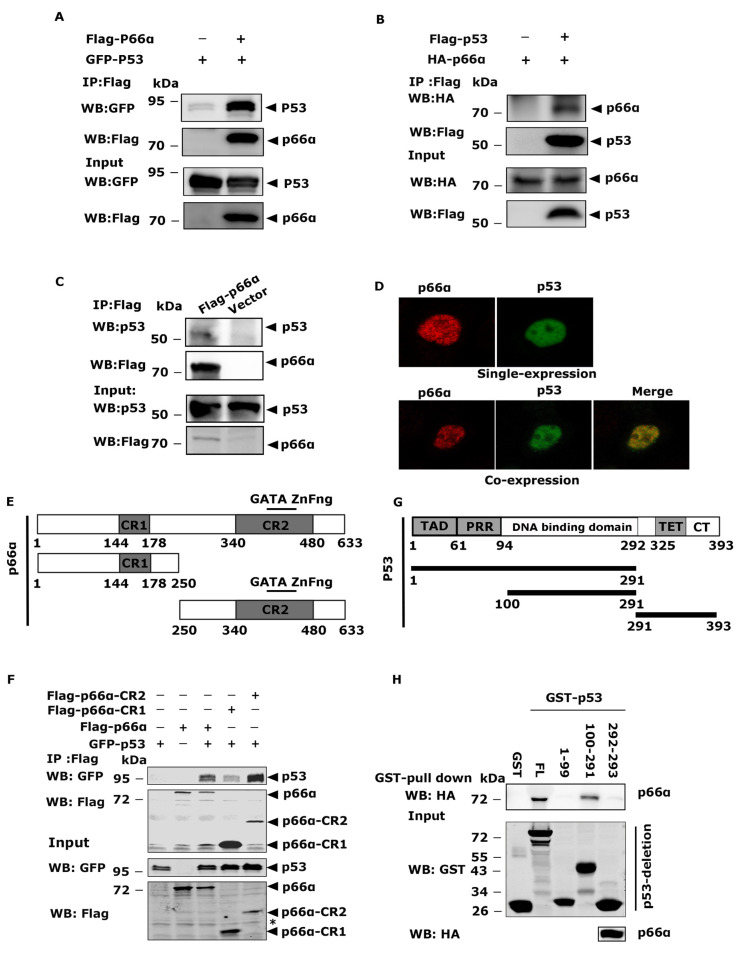
p66α directly binds the DNA-binding domain of p53. (**A**) The exogenously expressed Flag-p66α and GFP-p53 proteins interacted in HEK-293T cells. Immunoprecipitation was performed by using Flag M2 beads to enrich p66α, and Western blotting was probed with the anti-GFP antibody. (**B**) The exogenously expressed Flag-p53 and HA-p66α proteins interacted in HEK-293T cells. Immunoprecipitation was performed with Flag M2 beads to enrich p53, Western blotting was probed with the anti-HA antibody. (**C**) The exogenously expressed Flag-p66α and endogenous p53 proteins interacted in HEK293T cells. Immunoprecipitation was performed with Flag M2 beads, and Western blot was performed with monoclonal antip53 antibody. (**D**) Subcellular localization of p66α and p53 in HeLa cells. The plasmids encoding Flag-p66α and GFP-p53 were transiently transfected into HeLa cells, and the immunofluorescent images were taken with confocal microscopy. (**E**) Diagrams of full-length p66α and p66α deletion mutants. The CR1 and CR2 are the highly conserved regions in p66α. The CR2 domain contains a GATA-type zinc finger motif. (**F**) p53 mainly interacted with the CR2 domain of p66α, but there was also a weak interaction between p53 and CR1 domain. The asterisk “*” means nonspecific bands. (**G**) Diagrams of full-length p53 and p53 deletion mutants. (**H**) The region between 100 and 291 amino acid residues of p53 bound p66α. In vitro translated p66α protein was subjected to GST pulldown assays using the purified GST fusion p53 proteins.

**Figure 4 cells-10-03593-f004:**
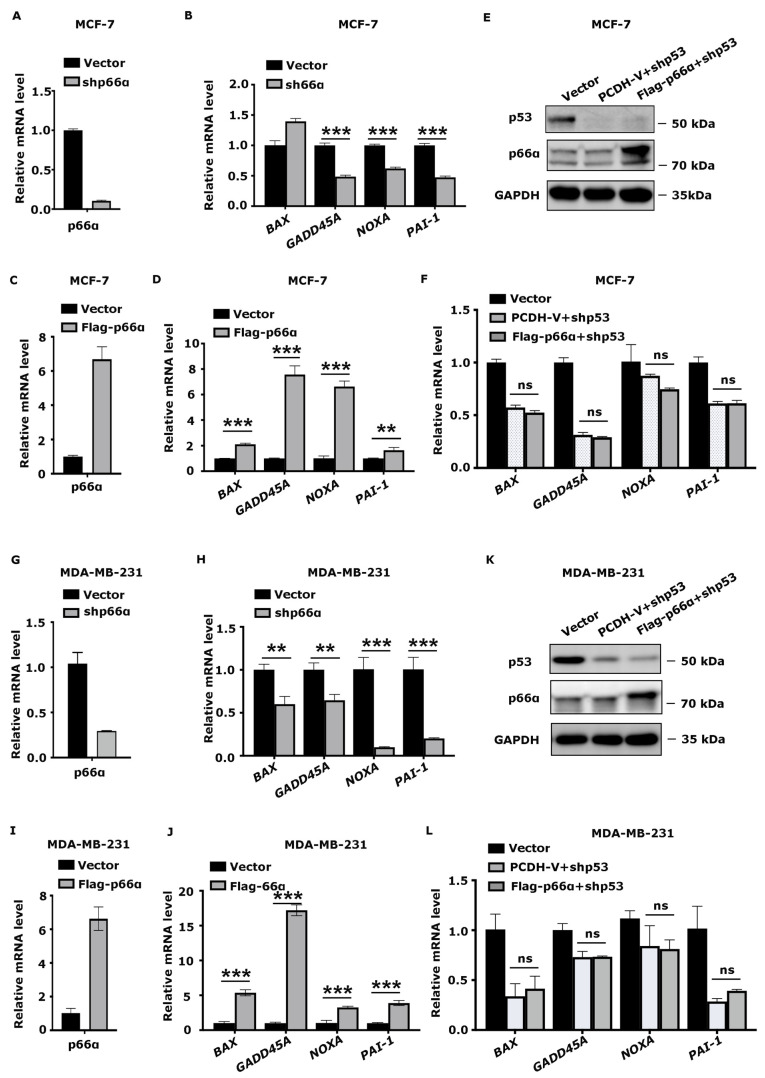
p66α promotes the expression of transactivated p53 target genes via p53. (**A**) qRT-PCR assays verified the mRNA level of p66α in MCF-7 shVector and shp66α cells. (**B**) The repression of p66α decreased the mRNA level of p53 target genes in MCF-7 cells. *** *p* < 0.001. (**C**) qRT-PCR assays verified the mRNA level of p66α in Flag-p66α overexpressing MCF-7 cells. (**D**) Overexpression of p66α increased the mRNA levels of p53 downstream genes in MCF-7 cells. ** *p* < 0.01, *** *p* < 0.001. (**E**) Western blot assays verified the protein levels of p53, p66α in p66α overexpressing MCF-7 cells after the depression of p53. (**F**) qRT-PCR assays of the mRNA level of p53 downstream genes after the depression of p53 in p66α overexpressing MCF-7 cells. “ns” indicates no significant difference (*p* > 0.05). (**G**) qRT-PCR assays verified the mRNA level of p66α in MDA-MB-231 shVector and shp66α cells. (**H**) Repression of p66α decreased the mRNA level of p53 downstream genes in MDA-MB-231 cells. ** *p* < 0.01, *** *p* < 0.001. (**I**) qRT-PCR assays verified the mRNA level of p66α in MDA-MB-231 Flag-p66α overexpressing cells. (**J**) Overexpressing p66α induced the mRNA level of p53 downstream genes. *** *p* < 0.001. (**K**) Western blot assays verified the protein levels of p53, p66α in p66α overexpressing MDA-MB-231cells after the depression of p53. (**L**) qRT-PCR assays of the mRNA level of p53 downstream genes after the depression of p53 in p66α overexpressing MDA-MB-231 cells. “ns” indicates no significant difference (*p* > 0.05). All data were shown as mean ± S.D. from three independent experiments.

**Figure 5 cells-10-03593-f005:**
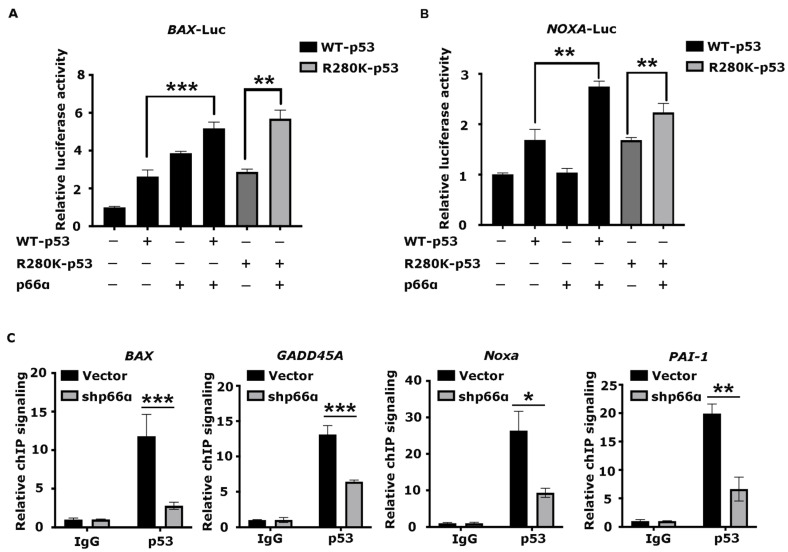
p66α promotes the transactivity of p53 by enhancing p53 binding at target promoters (**A**) Luciferase assays were performed in HEK-293T cells knocking down of p53. The value was normalized to β-gal. Both p53WT and p53R280K synergistically worked with p66α to transactivate the *BAX* promoter. (**B**) Luciferase assays showed that both p53WT and p53R280K synergistically worked with p66α to transactivate the *NOXA* promoter. (**C**) ChIP assays were performed in MDA-MB-231-shp66α and shVector cells using an antibody specific to p53. The enriched DNA fragments were amplified by qRT-PCR assays. * *p* < 0.05, ** *p* < 0.01, *** *p* < 0.001. All data were shown as mean ± S.D. from three independent experiments.

**Figure 6 cells-10-03593-f006:**
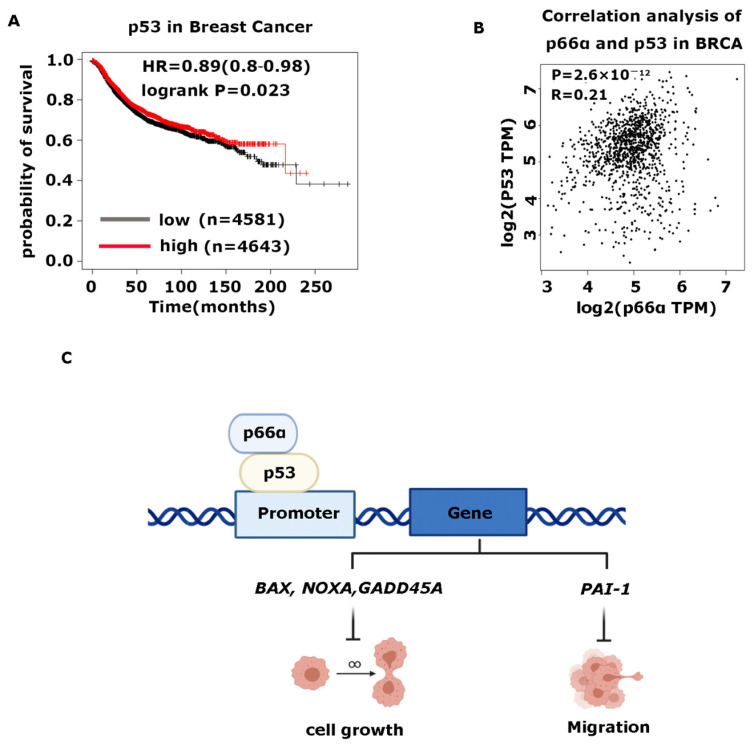
High expression of p66α and p53 is positively correlated with and predicts a good prognosis in breast cancer. (**A**) Kaplan-Meier plot of the relapse-free survival of patients with breast cancer in whole data sets stratified by p53 expression. (**B**) Scatter plot showed that p66α and p53 were positively correlated in breast cancer. The *p* value was calculated via Pearson’s ranking correlation coefficient analysis. BRCA: breast invasive carcinoma (TCGA database). TPM: Transcripts Per Kilobase of exon per Million mapped reads. (**C**) Schematic model for p66α mediated p53 transactivating genes expression. p66α interacted with p53 and promoted the transcriptional activation ability of p53 by enhancing p53 binding in target promoters, including genes that control cell growth, such as *BAX, GADD45A,* and *NOXA*, and that control cell migration such as *PAI-1*.

## Data Availability

All data generated or analyzed during this study are included in this published article.

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
