# Peer review of "p66α Suppresses Breast Cancer Cell Growth and Migration by Acting as Co-Activator of p53"

_cells, 2021, doi:10.3390/cells10123593_

Round 1

Reviewer 1 Report

In this manuscript Zhang et al, show that the GATA zinc finger domain-containing transcription factor p66a acts as a tumor suppressor in breast cancer. The authors show that lower levels of p66a in breast cancer patients correlate with poor prognosis. Likewise, KD of p66a increases cancer cell growth and migration in vitro and results in increased tumor formation and metastasis of breast cancer cells in mice. In addition, p66a is shown to associate with p53 via the DBD of p53. Additionally, the authors demonstrate that depletion of p66a decreases p53 target genes which can be rescued by ectopic expression of p66a. Further, the authors conclude that the binding of p53 to its target gene promoters is dependent on p66a. Overall the findings in the manuscript seem to be important and robust. However, the cell line used by the authors in the study (MDA-MB-231) is not ideal to study p53 function as p53 is mutated in this cell line. Here are some major and minor concerns with the manuscript.

Major concerns:

  • The major flaw of the study seems to be the use of MDA-MB-231 cell line to study the effect of p66a on p53 transcriptional activity. MDA-MB-231 cell line contains mutant p53 (R280K) which has been clearly shown to inhibit its binding to DNA and thus p53 in this cell line should be unable to regulate its target genes. Could the authors please explain their use of this cell line to study function of wild-type p53? Here are some of the references to show that the R280K mutant of p53 is defective in DNA binding and in activating its target genes:
    1. Malcikova, Jitka, Tichy, Boris, Damborsky, Jiri, Kabathova, Jitka, Trbusek, Martin, Mayer, Jiri and Pospisilova, Sarka. "Analysis of the DNA-binding activity of p53 mutants using functional protein microarrays and its relationship to transcriptional activation" Biological Chemistry, vol. 391, no. 2-3, 2010, pp. 197-205.
    2. Gomes, A.S.; Trovão, F.; Andrade Pinheiro, B.; Freire, F.; Gomes, S.; Oliveira, C.; Domingues, L.; Romão, M.J.; Saraiva, L.; Carvalho, A.L. The Crystal Structure of the R280K Mutant of Human p53 Explains the Loss of DNA Binding. J. Mol. Sci. 2018, 19, 1184. https://doi.org/10.3390/ijms19041184
  • In Results section 3.3, the authors mention that p66a “interacts” with p53 using co-immunoprecipitation experiments. However, co-immunoprecipitation only shows “association” between two proteins and not interaction, as both proteins may associate indirectly via some other protein(s).
  • Again, in Result section 3.4, the authors use GST-pull down assays to conclude that p66a directly interacts with the DBD of p53. Although the authors have used bacterially purified GST-p53, the HA-p66a protein is obtained from HEK-293T cells and is not pure. Thus, the authors cannot conclude a direct interaction of p66a with the DBD of p53.
  • In Figure 4, the effect of p66a KD/overexpression on p53 target genes is surprising given the fact that p53 is mutated and unable to bind to its target RE sites. Does this mean that p66a can regulate several p53 target genes independent of p53? However, there is not enough evidence provided by the authors to conclude this either. More surprising are the results from ChIP assays showing p53 binding to its target gene promoters, when in fact it should be unable to bind DNA.
  • The authors haven’t shown if p66a can itself regulate p53 protein/mRNA levels. Can the authors examine the effect of p66a KD/overexpression on p53 protein/mRNA?
  • In Figure 5, the authors' conclusion that high expression of p66a and p53 predicts good prognosis in breast cancer does not appear very robust. The number of patients analyzed for p66a expression in Figure 1A and for p53 expression levels in Figure 5 are not equal.

Minor concerns:

  • p53 blot is missing in Figure 3B.

Author Response

Dear Reviewer:

    We really appreciate the time and effort that you dedicated to providing feedback on our manuscript and are grateful for the insightful comments on and valuable improvements to our paper. We have worked very diligently to address these questions, and each comment has been thoroughly addressed as much as we can. We look forward to meeting your expectations.

Reviewer 2 Report

The authors explore the role of p66 in breast cancer models. Through silencing and overexpression experiments in breast cancer cell lines they find that p66a seems to act as a tumor suppressor restraining cell growth and migration ability. Consistently, p66a silencing increases the volume of breast xenografts and their ability to form lung metastases in mouse models. Also, from public databases it seems that higher p66 expression predicts better patient survival. The authors then investigate by mass spec possible p66 binding partners and identify p53. They confirm the interaction by reciprocal co-IPs and pull down experiments and show that both proteins localize to the nucleus. Finally, they show that p66a expression impacts on the ability of p53 to activate various p53 transcriptional targets including genes involved in apoptosis, cell growth and metastasis. Overall, the manuscript reports a relevant new protein interaction and disclose a new potential role for p66 in breast tumorigenesis that could have translational implications. I have a few suggestions as listed below:

Major:

The authors state that ‘these findings demonstrate that p66α is a tumor suppressor by functioning as a co-activator of p53’. However, I would tone down this sentence: can we really expect that p66a exerts all its antitumor potential through the action of p53? The authors could easily assess this by testing the effects of p66a modulation in a breast cancer cell line in which p53 is depleted.

Minor:

Abstract line 13: change to ‘essential for gene silencing’

Abstract line 14: change to ‘wide spectrum’

The abstract and the whole text require English editing.

Intro line 28: define MDB

Intro line 57: replace with ‘histone marks’

Tables 1 and 2 reporting PCR primers could be moved to supplementary material.

M&M 2.7. Cell invasion assay section: it seems that the authors report migration experiments, not invasion. Please amend.

M&M line 165: meaning reactants?

M&M line 167: change to: cells were stained

Results line 205: change to ‘To examine’

The authors should add details in the M&M section of the spectrometry assay following FLAG-p66a immunoprecipitation. Also, a more thorough description of the assay results in the ‘results’ section would be useful. Did they retrieve also already known p66 partners?

Please explain what the asterisk in figure 3F represents

Figure 5 A: please clarify if BRCA stands for BREAST CANCER (it could be confused with the BRCA gene)

Human genes should be written in capital and italics

Discussion lines 382-383: “the over-expression of p66α apparently inhibited cell growth and cell metastasis” please change metastasis with migration. These are normal breast cells, not supposed to have metastatic potential.

The addition of a schematic model is a nice idea but it seems too simple. Could the authors elaborate on this?

Author Response

Dear reviewer:

We really appreciate the time and effort that you dedicated to providing feedback on our manuscript and are grateful for the insightful comments on and valuable improvements to our paper. We have worked very diligently to address these questions, and each comment has been thoroughly addressed as much as we can. We look forward to meeting your expectations.

Reviewer 3 Report

Authors: Qun Zhang, Yi-Hong Zhang, Jie Zhang, Dan Zhang, Meng-Ying Li, Hui Zhang, Jia-Min Wang, Zhao-Yuan Hou, Xiu-Qun Zou   Title: p66α suppresses breast cancer cell growth and metastasis by acting as co-activator of p53    COMMENTS:  This is an interesting study that has been performed on an adequate model. The submitted manuscript is well written and nicely illustrated. The Authors have revealed a novel mechanism of the regulation of p53 and studied a role of this mechanism in breast tumor growth/metastases. Their findings indicate new molecular targets to treat breast cancer. I think that the submitted manuscript may be accepted in the present form. I would solely advise to mention mortalin (GRP75) as another regulator of p53 and discuss a possible interplay between the mortalin- and p66alpha-mediated pathways in regulations of p53.      

Author Response

Dear reviewer:

We really appreciate the time and effort that you dedicated to providing feedback on our manuscript and are grateful for the insightful comments on and valuable improvements to our paper.  The comments have been addressed as much as we can. We look forward to meeting your expectations.

Reviewer 4 Report

In their manuscript, Zhang et al. argue the role of p66α in the suppression of breast cancer growth and metastasis through interaction with p53 oncosuppressor.

This paper is interesting however some points need to be clarified.

The Materials and Methods should be better detailed:

-Para 2.2: Explain, in a better way, how the transient and stable transfection was performed.

-Para 2.7: For cell invasion assay, the authors stated that the number of migrated cells was counted, but how was this count performed (per field, at 200X …), and how many fields were counted? Report these data also in the histograms of figure 2 panels B and D.

-Para 2.8 and 2.9: It is unclear whether the mice were treated for 21 days or if they were analyzed 21 days after inoculation.

-Para 2.10: I do not understand this paragraph. Please explain this concept further.

The standard abbreviation for Haematoxylin and Eosin is H&E.

-Figure 2:

In Figure 2B and 2D in the histogram indicate on the Y-axis "Number of migrated cells/field", or explain how the count was performed.

In Figure 2E, indicate at what time the mouse image shown was recorded.

In Figure 2F, align the numbers on the X-axis

In figure 2H also show a lower magnification image (i.e. 2X or 4X) in which it is easy to identify the difference in metastatic foci between shp66α and sh-Vector.

The discussion needs to be rewritten; it should contain only the critical discussion of the results and not the results again.

Some sentences are difficult to understand, and grammatical errors in the manuscript require further correction. In my opinion, a revision of English would be appropriate.

Author Response

(The authors gave the same response as above.)

Round 2

Reviewer 1 Report

I am satisfied by the response provided by the authors and the manuscript seems better now. 

Reviewer 4 Report

The authors have made reasonable efforts to address my concerns. In my opinion, the paper should be accepted in its revised form.

Extensive editing of the English language and style is required.